# The Role of Mesenchymal Stem Cells in Modulating Adaptive Immune Responses in Multiple Sclerosis

**DOI:** 10.3390/cells13181556

**Published:** 2024-09-16

**Authors:** Sepehr Dadfar, Esmaeil Yazdanpanah, Alireza Pazoki, Mohammad Hossein Nemati, Majid Eslami, Dariush Haghmorad, Valentyn Oksenych

**Affiliations:** 1Student Research Committee, Semnan University of Medical Sciences, Semnan 35147-99442, Iran; 2Department of Immunology, School of Medicine, Semnan University of Medical Sciences, Semnan 35147-99442, Iran; 3Cancer Research Center, Semnan University of Medical Sciences, Semnan 35147-99442, Iran; 4Department of Bacteriology and Virology, Semnan University of Medical Sciences, Semnan 35147-99442, Iran; 5Broegelmann Research Laboratory, Department of Clinical Science, University of Bergen, 5020 Bergen, Norway

**Keywords:** multiple sclerosis, MS, central nervous system, CNS, mesenchymal stem cells, MSC, immunomodulatory properties, therapeutic potential

## Abstract

Multiple sclerosis (MS) is a chronic autoimmune disease of the central nervous system, leading to significant disability through neurodegeneration. Despite advances in the understanding of MS pathophysiology, effective treatments remain limited. Mesenchymal stem cells (MSCs) have gained attention as a potential therapeutic option due to their immunomodulatory and regenerative properties. This review examines MS pathogenesis, emphasizing the role of immune cells, particularly T cells, in disease progression, and explores MSCs’ therapeutic potential. Although preclinical studies in animal models show MSC efficacy, challenges such as donor variability, culture conditions, migratory capacity, and immunological compatibility hinder widespread clinical adoption. Strategies like genetic modification, optimized delivery methods, and advanced manufacturing are critical to overcoming these obstacles. Further research is needed to validate MSCs’ clinical application in MS therapy.

## 1. Introduction

Multiple sclerosis (MS) is an inflammatory demyelinating disease of the central nervous system (CNS), specifically featuring an autoimmune response resulting in inflammation, widespread primary demyelination, and progressive neurodegeneration as well as disruption of the blood–brain barrier (BBB), consequently leading to neurological dysfunction and permanent disability [1,2]. The estimated number of individuals affected by MS worldwide has increased from 2.1 million in 2008 to 2.8 million in 2020, with a greater prevalence in females [3,4]. Given that MS disease mostly targets the young group at the age of working, and given the high economic costs of the disease, it imposes a significant socioeconomic burden on societies [5,6]. Although the precise etiopathogenesis of MS is complex and still remains elusive, evidence supports the idea that an extremely complicated and multifactorial interaction between genetic, lifestyle, and environmental risk factors are of the utmost importance for MS disease initiation or progression [1,7]. Epidemiological investigations have shown that the risk of developing MS significantly increases with a raising degree of kinship, suggesting some degree of genetic susceptibility [8]. Regardless of the underlying cause, there is growing consensus that the immune system is the main culprit in MS and its animal models [9]. Among them, dysregulation or overactivation of the immune response are potentially involved in the pathogenesis and development of MS, leading to the infiltration of immune cells into the CNS and, consequently, the formation of demyelinating plaques [9]. The most prominent pathological hallmark of MS is the presence of focal demyelination, known as plaques or lesions, characterized by variable degrees of gliosis and inflammation and relative axonal preservation in the white and gray matter of the CNS [10].

Substantial insights into molecular mechanisms underlying the pathogenesis of MS and the development of therapeutic targets have emerged from studies using experimental autoimmune encephalomyelitis (EAE) [11,12]. The most commonly used model for studying autoimmune-mediated myelin degradation in MS is EAE, in which the disease can be induced (active EAE) in susceptible animals by immunization with one of the myelin-derived antigens, such as proteolipid protein, myelin oligodendrocyte glycoprotein (MOG), or myelin basic protein (MBP), leading to an immune response targeting myelin in the CNS [12,13]. In addition, EAE can also be passively induced through the adoptive transfer of activated myelin-specific helper T cells by intravenous injection into naïve animals [11]. 

Despite enormous progress over the past decades, there is currently no definitive treatment, and current treatment options have limitations in clinical use owing to their undesired side effects [14,15,16]. Most of the available therapeutic approaches, which are based on the prescription of immunosuppressive and immune-modulating agents, are aimed primarily at preventing the recurrence of relapses and the accumulation of disability, slowing the progression of disability as well as attenuating the symptoms [14,15,16]. Of note, these therapy options, generally, are not capable of hindering the ongoing neuronal degeneration and disease progression due to the loss of myelin [17]. Considering the limitations of conventional MS drugs, there is an urgent need to identify more effective and safe alternative therapies [16]. MSCs have garnered significant attention from scientists and clinicians as a potentially effective therapeutic intervention in various degenerative and immune cell-associated diseases due to their self-renewal capacity, multilineage differentiation, tissue and organ regeneration, and lack of immunogenicity [18]. In addition to these regeneration-promoting properties, MSCs hold potent immunomodulatory and anti-inflammatory capacity by using various mechanisms, such as extracellular vesicles (EVs) production, growth factors, mitochondrial transfer, cell contact-dependent, and soluble factors [19]. Numerous randomized controlled trials (RCTs) have shown that MSC-based investigations have increased hope for the treatment of various inflammatory and autoimmune disorders, including MS, rheumatoid arthritis (RA), systemic lupus erythematosus (SLE), type 1 diabetes (T1D), inflammatory bowel disease (IBD), and Crohn’s disease (CD), with the main mechanisms being the regulation of immunity and inhibition of inflammation [20,21].

Considering the aforementioned immune-regulatory properties of MSCs, in this study, we primarily focus on the interplay between MSCs and different immune cell subsets and discuss recent advances in the use of MSCs as potential therapeutic applications in the treatment of human MS population and its animal models.

## 2. Pathophysiology of Multiple Sclerosis

There continues to be a rising interest in understanding the potential role of the immune response in neuroinflammatory diseases such as multiple sclerosis. MS is the most common idiopathic inflammatory and neurodegenerative disability of the CNS in young adults of presumed autoimmune etiology [22]. It is believed to be driven by an irreversible breakdown of immunological self-tolerance, an imbalance in the immunomodulatory network, and an aberrant activation of both the innate and the adaptive immune system [23]. Many defects in the functioning and regulatory mechanisms of the immune system have been detected in MS patients, but the concept that myelin-specific T cells play a central and fundamental role in generating CNS autoimmunity prevails [9].

During TCR activation in a particular cytokine milieu, naïve CD4^+^ T cells can differentiate into several functionally distinct subsets, including Th1, Th2, Treg, or Th17 effector cells, which play a critical role in mediating adaptive immunity to a variety of pathogens [24]. In EAE/MS, much evidence has been obtained in support of the concept that effector Th1 (IFN-γ)- and Th17 (IL-17)-related cytokines contribute to the inflammatory response and demyelinating lesions within the CNS, while Treg- and Th2-related cytokines have been associated with anti-inflammatory effects and the improvement of symptoms [25,26].

The major proinflammatory CD4 T-cell populations involved in the initiation and perpetuation of the disease are the autoreactive Th1 and Th17 cells [27]. Until recently, Th1 cells were thought to be the main pathogenic T cells in EAE and MS [23,28,29]. As major subsets of CD4^+^ Th cells, Th1 populations originate from stimulation with IL-12 and IFN-γ and secrete proinflammatory molecules such as IFN-γ, TNF-α, TNF-β, and IL-1β [23,28,29]. The role of Th1 lineages in MS is further supported by reports that mice lacking T-bet and signal transducer and activator of transcription 4 (STAT4), in which Th1 cell development is impaired, are completely protected from the disease [30]. Similarly, systemic administration of IFN-γ, the signature cytokine of Th1 cells, to patients with MS has been shown to aggravate the clinical disease [30,31]. It is further supported by findings that show that the adoptive transfer of IFN-γ-producing T-cell lines could induce EAE [32].

Th17 cells, another subset of CD4^+^ Th cells, have now been characterized as a unique CD4^+^ T lineage with distinct secretion of IL-17, IL-21, IL-9, IL-22, and TNF-α, of which IL-17 is the most important cytokine in Th17-mediated encephalopathy [23,33]. Recent data show that IL-6 and TGF-β are necessary during Th17 cell differentiation, while their survival and expansion are dependent on IL-21 and IL-23 [23,33]. More recently, Th17 cells have been increasingly considered to be the major driver of CNS inflammation based on evidence from animal models and patients with MS [23,34]. Several studies further confirmed the significant immunopathogenesis role of Th17 cells in MS and EAE. The frequencies of Th17 cells and IL-17 were found to increase in MS lesions [35], where they are thought to contribute to the disruption of the blood–brain barrier [33]. Moreover, infusion of Th17 cells or administration of IL-17 has been found to significantly exacerbate the severity of EAE clinical symptoms, while IL-17 or IL-17 receptor-deficient mice as well as IL-17-specific inhibition have been shown to attenuate EAE severity, indicating the central role of Th17 cells in proinflammatory responses in EAE [23,36,37].

The auto-reactive myelin-specific Th1 and Th17, which are activated in the peripheral lymph nodes, cross the blood–brain barrier (BBB) and enter the steady-state CNS, where T cells are reactivated by their lineage-defined cytokines, such as IFN-γ and IL-17, leading to demyelination, axonal damage, and spinal cord and brain lesions [38,39]. Interestingly, although both Th1 and Th17 effector cells are capable of inducing EAE, the pathological features and clinical signs may differ. IL-23-polarized T cells have been shown to promote the expression of neutrophils and granulocyte–colony-stimulating factors (CSFs), whereas macrophage-rich infiltrates and prominent NOS2 upregulation have been found to be prominent in IL–12–driven lesions [40]. In addition to these findings, IFN-γ producing Th1 subsets are required for the generation of classical EAE, whereas Th17-like cells producing IL-17-induce atypical EAE [28]. Therefore, IL-17 producing Th17 cells and IFN -γ producing Th1 cells may play complementary roles in the immunopathogenesis of EAE [28,40].

In contrast to the Th1 and Th17 subtypes, Th2 effector cells that are induced by IL-4 and produce IL-4, IL-5, IL-10, and IL-13, have been associated with inflammation reduction and improvement of symptoms in MS patients [41]. Accordingly, the IL-4-deficient C57BL/6 mice were found to be differently susceptible to EAE and developed a more severe form of the clinical disease [42], while IL-4 treatment resulted in amelioration of the clinical disease [43]. Furthermore, the clinical recovery from acute EAE appears to be associated with increased levels of IL-4 and IL-10 mRNA within the CNS [44]. Likewise, IL-4 have the ability to reduce pathological inflammation via an increase in M2 macrophages [45]. The neuroprotective effect of Th2 is also supported by the observation that treatment with the induction of MBP-specific Th2 cells alleviates demyelination and the suppression of the production of inflammatory cytokines in the CNS [43]. 

An intriguing relationship between the generation of inflammatory Th1 and anti-inflammatory Th2 subsets has recently been demonstrated. Th1 and Th2 responses are generally considered mutually antagonistic, as evidenced by the Th2-cytokine IL-4 and IL-10 strongly suppressing differentiation of naïve cells toward the Th1 pathway and vice versa, and the Th1-cytokines IL-12 and IFN-γ inhibit differentiation of naïve cells toward the Th2 differentiation [46,47,48] (Figure 1).

As master regulators of immune responses, CD4^+^ regulatory T cells (Tregs) expressing the FoxP3 transcription factor have the ability to modulate the function of effector T cells, maintaining immunological homeostasis and preventing autoimmunity, thereby serving a protective function in autoimmune diseases [24]. IL-10 and transforming growth factor-β (TGF-β), the main secreting productions of Treg, have been shown to limit pathological autoreactive immune responses and the cascade of proinflammatory cytokines activation [49]. Notably, decreased frequencies and functionally impaired Tregs cells have been detected in MS lesions, and have also been found to correlate with an exacerbation of disease symptoms [24]. Besides, several studies have described that the transfer of Treg cells significantly attenuates EAE symptoms or pathological severity, while Treg depletion has been reported to exacerbate EAE severity, accompanied by increased production of proinflammatory cytokines and proliferation of autoreactive effector T cells [50,51].

In addition to the abovementioned cells, over the last decade, research has shifted the focus of attention toward CD8^+^ T cells as evidence increasingly demonstrates that CD8^+^ T cells predominate over CD4^+^ T cells in CNS lesions of MS patients [52,53]. CD8^+^ T cells, also known as cytotoxic T lymphocytes (CTL), are characterized by their ability to secrete high levels of proinflammatory cytokines, such as TNFα and IFN-γ, and cytotoxic properties through perforin and granzyme and Fas pathways [54,55]. Cytotoxic T cells are implicated in the pathogenesis of MS during the relapse phases and in the chronic phase [45,56]. Under inflammatory conditions, most CNS resident cells including oligodendrocytes, astrocytes, and neurons express MHC class I antigens, making them potential targets for cytotoxic CD8^+^ T cells [52,57]. Intriguingly, the recent growing body of literature suggests that CD8^+^ T cells not only induce or aggravate tissue destruction in the CNS in murine and human autoimmune disorders, but also has regulatory properties and is involved in suppressing disease progression and/or severity by producing immunosuppressive cytokine IL-10 [52,54,58]. Therefore, CD8^+^ T cells can act as a double-edged sword and further research is needed to elucidate the roles of these subsets in MS.

Although much focus on MS pathology and its model animals has centered on autoreactive T cells, remarkable evidence from patients with MS points to a role played by B cells in disease pathogenesis in various ways, including antigen-presenting cells and production of proinflammatory cytokines and/or autoantibodies [9]. One of the most compelling findings highlighting the major role of B cells in the disease has been the B cell depletion efficiently suppressing inflammatory disease activity [23,59].

Since MS is driven by the excessive activity of autoreactive Th1 and T17 cells and is suppressed by Th2 and Treg cells, the development of novel therapeutic approaches based on manipulating immune cells, in particular T cell subtypes, can be very promising in the therapy of immune-mediated and inflammatory disorders such MS [25].

## 3. Mesenchymal Stem Cells: Characteristics and Mechanisms of Action

MSCs are a group of nonhematopoietic stem cells with high self-renewal ability that were initially isolated from bone marrow and subsequently identified in many other tissues, such as adipose tissue, epidermis, umbilical cord blood, and the periendothelial area [60,61]. They are characterized by their extensive capacity to differentiate into cells indicative of various mesenchymal lineages, such as osteoblasts, chondrocytes, fibroblasts, and adipocytes in response to appropriate stimuli [60,61]. According to the tissue of origin, MSCs can express a wide range of surface markers and cytokine profiles [62]. Phenotypically, the common characterization markers of MSCs are CD105, CD90, and CD73 and the absence of hematopoietic progenitor cells markers such as CD45, CD34, CD14, CD19, and HLA class II [62]. Owing to paracrine potentials, ease of accessibility and separation, long-term ex vivo proliferation, differentiation multipotency, and their anti-inflammatory action, they are considered a desirable cell source in regenerative medicine [63,64]. Recently, MSC-related cell therapy has been shown to be a promising therapeutic strategy in a broad panel of immune disorder-related diseases and degenerative diseases such as MS, mainly attributable to the neuroprotective and immunomodulatory effects of these cell. 

Intensive research has been undertaken to unravel the mechanisms by which MSCs mediate their immunosuppressive properties. An increasing number of studies have shown that immunoregulation by MSCs is mediated directly by two distinct mechanisms: interactions with immune cells through cell-to-cell contact and soluble factors, including nitric oxide (NO), hepatocyte growth factor (HGF), fibroblast growth factor (FGF), indoleamine-pyrrole 2,3-dioxygenase (IDO), PGE2, IL-4, IL-10, TNF-α, and TGF-β1, all of which are involved in the suppression of proinflammatory signaling and augmentation of regulatory networks [65,66].

The interplay between MSCs and immune cells is being investigated in numerous clinical trials. The MSCs exert an immunomodulatory function on both the innate and adaptive immune system such as T, B, NK cells, and DCs, making them beneficial for treating conditions caused by abnormal immune activation [67]. The evidence is increasingly indicating that MSCs are also capable of inducing immune cells with more anti-inflammatory or tolerant phenotypes [68], accompanied by a shift from pathogenic inflammatory Th1 and Th17 cell subtypes to anti-inflammatory Th2 and Treg cells [14,69,70,71]. In response to inflammatory mediators, MSCs undergo a process of selective migration to injury and inflammation using cell adhesion molecules and receptors of cytokines and subsequently stimulate potent immunomodulatory and anti-inflammatory activities [72]. Under inflammatory stimulants, MSCs are able to promote PD-L and FasL, which repress pathogenic CD4^+^ T cell responses and cause direct killing of activated T cells, respectively [73]. Consistent with these observations, a number of studies have suggested that MSCs repress the inflammation process, partly by downregulating the production of proinflammatory cytokines produced by Th1 and Th17 such as IL-17, IL-1β, IL-6, IL-21, IL-22, IFN-γ, TNF-α, and IL-12, while enhancing the production of anti-inflammatory mediators produced by Th2 and Treg cells including IL- 5, IL-4, IL-10, and TGF-β [65,67,74,75,76] (Figure 2).

Besides CD4^+^ helper T cells, CD8^+^ cytotoxic T cells are also susceptible to the suppressive activity of MSC. Several lines of evidence have demonstrated that MSCs could inhibit the proliferation and cytotoxic function of CTLs and modulate their cytokine production through TGF-β and IL-6 [67,72]. Most importantly, they can inhibit the differentiation of cytotoxic T lymphocyte precursor cells (CTL-P) into CTL effector cells through the secretion of suppressive factors [77]. 

The humoral immune system may be affected by the immunosuppressive properties of MSCs. It has been well documented that both human and murine MSCs could suppress B cell differentiation, maturation, and plasma cell differentiation, as well as antibody production [78,79,80]. Moreover, they promote the polarization of the B cell phenotype toward the B regulatory cells (Bregs) phenotype, which releases IL-10, possesses anti-inflammatory and immunoregulatory properties, and modulates the immune environment homeostasis [81].

MSCs are reported to inhibit the immune response by modulating antigen presentation by DCs. Growing evidence supports that MSCs not only suppress conventional DCs, but also induce regulatory DCs [67]. Strikingly, a number of studies have shown the capacity of MSCs to disrupt the maturation and function of DC through the downregulation of MHC class II molecules and costimulatory molecules, resulting in the production of the nonfunctional T cells [67,82,83]. In addition, MSCs have been shown to polarize macrophages from a proinflammatory M1 profile toward a neuroprotective and anti-inflammatory M2 profile through the NF-κB and STAT3 pathways and secrete soluble factors, leading to the suppression of proinflammatory cytokines secretion [84,85]. Furthermore, it has been demonstrated that MSCs are capable of suppressing NK-cell proliferation, cytokine secretion, and cytotoxicity through immunosuppressive secretors, such as PGE2, TGF-β, and sHLA-G [86]. 

In addition to these immunomodulatory functions, MSCs also have immunosupportive capacities. It is worth noting that MSCs have also been shown to attenuate microgliosis and astrogliosis, as well as demyelination, through the secretion of neuroprotective factors [14,87]. Another intriguing feature of MSCs is their ability to evade immune recognition and even actively inhibit immune responses. This hypoimmunogenic or immune-privileged property is attributable to the low expression of class II major histocompatibility complex (MHC-II) and costimulatory molecules such as CD40, CD80, and CD86 in their cell surface [70].

However, MSCs may exert differential functions according to certain mediators present in their microenvironment, as several conflicting findings have shown that MSCs can both fortify and restrain the inflammation under several conditions. In fact, the immunosuppressive ability of MSCs is dependent upon the types and strengths of the inflammatory signals they receive [80,88].

## 4. Preclinical Studies Investigating MSC Therapy in MS

### 4.1. Overview of Preclinical Studies Using Animal Models of MS

Several animal tests and clinical investigations have found stem cell treatment to cure or prevent CNS damage [89]. It has been demonstrated that bone-marrow-derived mesenchymal stem cells (BM-MSCs) have immunomodulatory and regenerative abilities [90]. Therefore, a novel treatment strategy for autoimmune diseases, such as MS, has been suggested: the transplantation of BM-MSCs in an EAE model [90]. Research has demonstrated that BM-MSCs can decrease the quantity of effector T cells in BM-MSC-transplanted EAE mice while increasing the amount of regulatory T cells (Treg) in the mice, inhibiting effector T cells and lessening EAE [90,91]. Furthermore, it has been proposed that BM-MSCs can drive macrophages in the spleen to polarize from the M1 type to the M2 type and reduce their production of TNF-α, a phenomenon which will relieve EAE [90]. BM-MSC-transplanted EAE mice have been found to express more anti-inflammatory cytokines like TGF-β and IL-10, while producing fewer inflammatory cytokines like IL-1β and IFN-γ [92] (Figure 3).

Our previous study’s findings suggest that exosomal miRNAs may be useful in treating EAE [93]. Delivery of BM-MSCs was demonstrated to diminish miR-21, miR-223, and miR-155 and increase miR-146, a phenomenon which was coupled with a decrease in Th1- and Th17-associated cytokines such as IFN-γ, IL-17, and IL-12 [93]. The study also showed that there was an increase in TGF-β and IL-10 expression [93]. Another study conducted by our group revealed that injecting synergic and allogeneic adipose-derived mesenchymal stem cells may improve the clinical score of EAE mice. This improvement was correlated with a decrease in the expression of inflammatory cytokines and an increase in anti-inflammatory cytokines in the spleen and central nervous system [94].

Studies have indicated that conditioned media produced from MSCs can contribute to the healing of EAE by combining immune inhibition and myelination induction [95]. Rajan and colleagues have shown that preconditioning neurons with human periodontal ligament mesenchymal stem cells (hPDLSC-CM) inhibits TLR-4 activation in the lipopolysaccharide (LPS)-activated mice motoneurons, preventing cell death linked to LPS activation [96]. Furthermore, it has been shown that the levels of neuronal growth factors and trophic factors Nestin, NGF, NFL-70, and GAP43 are elevated in the neurons preconditioned with hPDLSC-CM. This finding demonstrates that hPDLSC-CM plays a role in promoting neuronal growth [96]. Through two different routes, it has been demonstrated that MSCs may enhance the myelinating capacity and engraftment of allogenic oligodendrocyte progenitors in demyelinated mice. First, by producing anti-inflammatory cytokines that diminish microglia and astrocytosis, and, second, by inducing trophic support through the release of growth factors like IGF-1 that facilitate the development of progenitor cells of oligodendrocytes into myelinating oligodendrocytes [97]. 

Additionally, it has been shown that IFN-γ-primed human MSCs may suppress the expression of Th17-specific transcription factors STAT3 and ROR-γt while increasing the amount of Treg transcription factor Foxp3, which affects the Th17/Treg balance and reduces inflammation, therefore curing EAE in animal models [98].

### 4.2. Findings on the Effects of MSCs on Adaptive Immune Responses in MS

MSCs can modify T-cell responses, affecting the adaptive immune response during the pathogenesis of MS [99]. According to some research, human umbilical-cord-derived mesenchymal stem cells, or hUC-MSCs, can boost the expression of Th2 anti-inflammatory cytokines like IL-4 and IL-10 while suppressing the expression of Th1 inflammatory cytokines like TNF-α. This has an immunomodulatory effect on the immune system in MS patients [100]. hUC-MSCs have been demonstrated to increase the hepatocyte growth factor (HGF), a multifunctional cytokine that promotes tissue healing via epithelial cells and IFN-γ levels in MS patients [100].

HLA-DRB1, an important agent that increases during the disease and is closely related to disease progression, is downregulated in patients with multiple sclerosis following umbilical cord mesenchymal stem cell transplantation. In addition, patients have shown lowered expression of proinflammatory cytokines like IL-17c and IL-2 [101]. In another study, the immunomodulatory effects of MSCs in patients with progressive multiple sclerosis were indicated by the induction of CD4^+^CD25^+^FoxP3^+^ regulatory T-cells and the diminishment of lymphocyte proliferation and CD3^+^CD69^+^ and Lin-CD11c^+^CD86^+^cells population, as activated lymphocytes and antigen-presenting cell populations [102].

Like MSCs, autologous mesenchymal stem-cell-derived neural progenitors (MSC-NPs) have been shown, by Violaine et al., to suppress immune responses in the central nervous system (CNS) and inhibit T-cell proliferation through the induction of Treg differentiation from naïve T CD4^+^ and the augmentation of HGf, IL-10, and indoleamine-2,3-dioxygenase (IDO) [103]. It has also been shown that MSC-NPs increase TLR-2 expression, which affects MSC differentiation and proliferation and the immunomodulatory effectiveness of MSCs [103].

## 5. Clinical Trials Evaluating MSC Therapy for MS

### 5.1. Overview of Clinical Trials Using MSCs in MS Patients

Conventional treatments for MS focus on symptom management rather than addressing the disease’s root cause [104]. However, recent research has shifted toward innovative therapeutic approaches, including stem cell transplantation. In vitro studies, animal experiments, and clinical trials have explored the potential of mesenchymal stem cells (MSCs) in MS treatment, aiming to enhance our understanding of disease mechanisms and develop more effective interventions [14].

As mentioned, MSCs can be derived from various tissues. In clinical trials, three main types are commonly used: those derived from bone marrow (BM-MSCs), adipose tissue (AD-MSCs), and umbilical cord (UC-MSCs). BM-MSCs have been the first to be discovered and investigated. However, subsequent research has found that AD-MSCs and UC-MSCs show better efficacy. These alternative sources have been later incorporated into clinical studies [105]. The dosage of MSCs for injection varies based on the patient’s condition and typically ranges between 2 × 10^5^ and 2 × 10^6^ cells per kilogram, administered either intravenously or intrathecally [14,106].

Cohen et al. conducted an open-label, phase 1 study (NCT00813969) to assess the feasibility, safety, and tolerance of autologous MSC transplantation. In this study, 26 patients with relapsing–remitting multiple sclerosis (RRMS) and secondary progressive multiple sclerosis (SPMS) received intravenous injections of 1–2 × 10^6^ cells per kilogram. The results confirmed the safety, feasibility, and good tolerability of this treatment approach, with no adverse effects or disease activity observed during the 6-month follow-up period [106].

In an open-label, phase 1 and phase 2 study (NCT0136424), Lu et al. investigated the safety and efficacy of simultaneous intravenous and intrathecal injection of MSCs over a 10-year follow-up period. The combined approach did not lead to any serious side effects and resulted in improvements in the expanded disability status scale (EDSS) index and overall patient conditions. Only minor side effects, such as headache and fatigue, were observed in the patients [107].

In a pilot study conducted by Harris et al. (NCT01933802), patients who received two-to-five intrathecal injections of MSC-derived neural progenitors (MSC-NP) were followed up for 7 years. The method was well tolerated, with only minor adverse effects such as fever and headache reported. Notably, MS did not progress in these patients and some individuals showed improvement based on the EDSS index; these positive results were observed when patients received at least 2 × 10^6^ cells per kilogram injections [108,109]. Table 1 displays a summary of clinical trial information from clinicaltrials.gov regarding the treatment of MS patients through MSC transplantation.

### 5.2. Safety and Efficacy of MSC Therapy in MS Patients

The safety and efficacy of MSC therapy have been confirmed through clinical trials and animal experiments. A systematic review and meta-analysis study revealed that, according to the EDSS index, an essential measure for assessing disability and other MS-related issues, 40.4% of patients showed improvement, 32.8% remained stable, and 18.1% experienced worsening of their conditions [15].

In this therapeutic method, there were no dangerous or serious adverse effects observed. Only minor effects were reported, including headaches in 57% of patients, fever in 53%, urinary infections in 23.9%, and respiratory tract infections in 7.9%. In terms of the injection route, two main types are intravenous (IV) and intrathecal (IT). When observing recovery, the IV route showed a 57% success rate, while the IT route had a 32% success rate. In terms of cell type, there are three primary sources of MSCs: bone marrow, adipose tissue, and umbilical cord. Among these, UC-MSCs exhibit greater effectiveness compared to BM-MSCs. Additionally, due to their lower immunogenicity, UC-MSCs have a reduced risk of transplant rejection. However, it is important to note that BM-MSCs remain more stable than UC-MSCs [15].

Due to clinical trials, this treatment method has limitations. Firstly, the specific injection dose remains uncertain, varying based on individual patient conditions. Additionally, the allogeneic MSC type raises concerns about transplant rejection, and a definitive therapeutic window has not been established. Addressing these challenges will necessitate large-scale clinical trials [15].

## 6. Challenges and Future Directions

MSC therapies have been the subject of more than 300 clinical trials, involving patients with autoimmune and degenerative conditions among others. Although MSC therapies have made enormous advances in recent decades, several challenges remain to be overcome.

MSCs have received regulatory approval in certain countries due to their generally acceptable safety profile and potential therapeutic effects in specific clinical contexts. However, with studies reporting inconsistent results, the true safety and clinical efficacy of MSC therapy have yet to be demonstrated, and a better knowledge of limits could lead to broader clinical implementation of safe cell therapies while avoiding damaging strategies.

### 6.1. The Effectiveness and Heterogeneity of Various MSC Populations

Many factors, such as donors and tissue sources, cell populations, culture settings, cell separation methods, and cryoprotective and thawing processes, all influence the heterogeneity of MSCs [110]. The International Society for Cell Therapy (2006) defined MSCs as a plastic adherent population that can differentiate down mesodermal lineages, undergo clonal expansion, and express stromal markers CD73, CD90, and CD105 while expressing none of the following: CD19/20, CD34, or CD45 [111]. However, these markers are not unique to MSCs; they are also found in other stromal cells. This suggests that this criterion is too broad to accurately define a specific population for therapeutic purposes. Therefore, MSCs will need to be more rigorously characterized and purified in the future before they can be approved as therapeutic interventions.

MSCs collected from multiple donors have distinct functions because of differences in age, health state, and other individual characteristics. Furthermore, MSCs derived from various tissues, such as adipose tissue and bone marrow, may exhibit differences in surface markers and abilities to differentiate. This variance likely arises from distinct biological, chemical, and mechanical stresses in stem cell environments, despite the same circumstances in laboratory cultures [110].

The age of the donor appears to be the most crucial consideration, hence why autologous transplantation may have certain restrictions. Younger donor cells proliferate more quickly, aging significantly more slowly in culture, and are less vulnerable to oxidative damage and alterations [112,113]. The challenging problem is how to grow MSCs from senior citizens in order to produce a sufficient quantity of therapeutic cells. Additionally, isolating a viable population of MSCs from patients with conditions such as diabetes, rheumatoid arthritis, and other inflammatory diseases is challenging. This difficulty, due to the potential impact of the disease on the cells, leads researchers to suggest that these autologous cells may lose their therapeutic efficacy [114,115].

Furthermore, there is compelling evidence suggesting that the circumstances under which MSCs are grown outside of the body might influence their physical characteristics and, potentially, their role. Some examples of these modifications include alterations in the physical structure, size of cells, ability to specialize, and the presence of certain proteins on the cell surface, such as CD44, CD105, CD146, and CD271 [116]. Moreover, there can be notable differences in both cell shape and differentiation ability within a single clone. For instance, Andrzejewska and colleagues showed that cells found at the outer edge express higher levels of genes associated with cell proliferation (MKI67 and PODXL), whereas genes related to the extracellular matrix (VCAM1) are more commonly expressed in interior MSCs [117].

Because there are currently no functional markers that can be utilized to differentiate MSC from fibroblasts or other MSC-like stromal cells, researchers have been persistently investigating distinct cell surface markers and molecular signatures to pinpoint specific cell subsets within heterogeneous MSC populations. Microarray profiling and RNA sequencing have shown transcriptional signals that indicate the possibility of differentiation. The primary transcription factors involved in osteoblast differentiation are osterix and distal-less homeobox5, whereas the capacity for adipogenic growth is linked to peroxisome proliferator-activated receptor gamma (PPAR-γ) and CCAAT/enhancer-binding protein alpha [118]. A single-cell-derived colony consisting of rapidly proliferating cells demonstrates high efficiency in colony formation. The cell surface markers STRO-1, CD146, and CD271 have been associated with this particular subset. Nevertheless, even cell subsets that share these surface markers can display varying chondrogenic differentiation potentials despite being cultured under identical conditions [119,120]. Moreover, MSCs with distinct surface markers associated with their differentiation capacity can demonstrate a range of physiological roles. While CD106+ MSCs have exhibited improved immunosuppressive properties and higher multipotency, CD105+ MSCs have shown myogenic potential, supporting the regeneration of infarcted myocardium [121,122,123].

Despite the significant promise of MSCs for therapeutic applications, several challenges remain. One of the key obstacles to their clinical use is the inherent heterogeneity of MSC populations. Increasing evidence shows that MSCs comprise multiple subsets with specific surface markers. Therefore, more work is needed to define these subpopulations based on biomarkers and biological functions.

### 6.2. Migratory and Homing Capacity of MSCs

The therapeutic efficacy of MSCs relies on their capacity to migrate to the injury site, adhere, and integrate into the target tissue. The direction of MSC migration is governed by the expression of chemokine receptors on MSCs and the presence of chemokines within the tissues [124]. The homing ability of MSCs is influenced by multiple factors. Various factors, including host receptivity, delivery technique, donor age, number of passages, and culture conditions, are crucial [125]. 

Ries et al. demonstrated that freshly isolated cells exhibit higher engraftment efficiency compared to cells cultured in vitro [126]. This discrepancy may be attributed to the aging and differentiation processes that cells undergo under in vitro culture conditions [127,128]. Culture conditions also play a major role in homing capacity, since they can alter the surface markers that are expressed differently during this process. For instance, the chemokine receptor CXCR4 is abundantly expressed on primary bone marrow MSCs, but its expression diminishes with successive passages, leading to reduced recognition of its ligand, CXCL12 [129,130].

In addition, chemokine expression profiles in injured tissues often differ from MSC receptor expression profiles. For example, in infarcted myocardium, the levels of CXCL1, CXCL2, and CCL7 are elevated, whereas the expression of their corresponding receptors, CCR1 and CXCR2, on MSCs is minimal. This disparity leads to a reduced efficiency in MSC migration to the infarct sites [131]. MSCs can be genetically engineered to express specific chemokine receptors, thereby enhancing their migration capabilities. For example, CCR7-modified MSCs have demonstrated clinical efficacy in mouse models of GVHD, while CXCR5-modified MSCs have been found to effectively migrate to injured sites by binding to CXCL13, which is highly expressed in damaged tissues [132,133,134].

In summary, primary MSCs are anticipated to exhibit superior therapeutic efficacy due to their enhanced migration capabilities. Moreover, genetically modified MSCs offer a unique treatment modality, providing the potential for targeted therapies.

### 6.3. Route of Administration

The route of delivery has a major impact on the effectiveness of cell therapy. Differences in delivery methods may have an impact on paracrine function, cell homing, and survival. The method that is most frequently used to administer MSCs is intravenous infusion [125]. Systemic infusion of MSCs has been demonstrated to exhibit notably low homing effectiveness, with the majority of research indicating that MSCs primarily become entrapped in the lungs and liver [111]. For multiple sclerosis, administering BM-MSCs intravenously has proved ineffective, as studies in mouse models of MS have shown that the mouse MSCs fail to reach the central nervous system’s inflammatory sites. Instead, the cells primarily accumulate in the lungs and liver [135]. Many studies have demonstrated that intratissue or intraorgan administration is more efficient and has a higher delivery retention than intravenous delivery [136]. For example, Augello and colleagues demonstrated that site-specific injection of MSC may be more effective than systemic infusion in RA [137]. Thus, to optimize therapeutic efficacy, it is essential to consider both the migratory capacity of MSCs and the selection of appropriate delivery methods.

### 6.4. Immunological Compatibility of Mesenchymal Stem Cells

MSCs have demonstrated low immunogenicity in vitro due to their limited expression of MHC I molecules, absence of MHC II expression, and lack of costimulatory molecules such as CD40, CD80, and CD86. MSCs have a low rejection rate when transplanted as allogeneic cells. In general, it is thought that the original MSCs have little immunogenicity [138]. Because most MSC products are produced by amplifying a limited number of donor cells, improper procedures and culture conditions may increase the immunogenicity of MSCs. When xenogenic MSCs, such as human MSCs, are administered to animal models, the in vivo inflammatory molecules after MSC infusion further reduce MSC viability and differentiation capacity, while also increasing MSC immunogenicity [110]. According to some studies, TGF-β reduces MHC-II expression in MSCs, while inflammatory substances including interferon-γ, higher cell density, and/or serum deprivation can cause high expression of MHCII in MSCs [139]. In addition, most studies suggest that a single MSC transplant is safe and does not cause an immunological response. However, in the event of long-term treatments involving repeated infusions, immune compatibility between donors and recipients is crucial to lower the risk of rejection. It has been observed that repeated intra-articular injections of allogeneic MSCs are more likely to result in an adverse reaction than autologous cells when administered in the same way [140,141]. 

Therefore, gaining insights into the molecular and cellular mechanisms responsible for the immune incompatibility of MSCs will aid the enhancement of the production of MSC-based therapies, as ensuring high-quality MSCs is crucial for the safety and effectiveness of clinical trials.

### 6.5. Restricted Proliferation of MSCs

In order to assess the effectiveness of a particular MSC population, it is necessary to provide MSCs of identical origin to the patients. Consequently, to produce an adequate number of cells, MSC expansion through large-scale ex vivo culture is frequently needed. In theory, MSCs can be cultured and expanded indefinitely in traditional plates and flasks to meet experimental needs. However, as the culture duration extends and passage numbers increase, MSCs encounter the Hayflick limit, leading to a significant reduction in proliferation. Additionally, their morphology changes from a thin spindle shape to a flattened square shape. For example, compared to fetal MSC populations, which maintain their immunosuppressive capacities over an extended culture period (~25 passages), adult BM-MSC lose their capacity to inhibit the immune system quite early (passages 5–7) [142,143].

MSCs’ stem cell properties are negatively impacted by the long-term, large-scale growth of these cells in 2D plates. In an aGVHD animal model, Zhao et al. discovered that hUC-MSCs, at different passages, display numerous changes in signatures and functions, with high-passage cells displaying diminished therapeutic benefits [144]. Furthermore, compared to MSCs in 3D cultures, those in 2D cultures are less effective at chondrogenic differentiation [145]. As a result, 3D expansion techniques have been created to preserve the phenotypic integrity of MSCs and avoid the flattened and wide morphology observed in monolayer cultures.

Together, these factors indicate that the development of a method that can quickly and affordably produce large numbers of cells with guaranteed cell quality is essential for the advancement of MSCs in clinical practice. It is also necessary to clarify the impact of ex vivo cultures on the integrity of MSCs and their functionality.

### 6.6. Long-Term Side Effects of MSCs Therapy

The safety of MSC-based therapies has been the subject of numerous studies to date. Karussis et al. investigated the effects and safety of MSC therapy in individuals with amyotrophic lateral sclerosis and multiple sclerosis. Over the course of a 25-month study including 34 participants, no significant side effects from the medication were observed. Furthermore, MRI scans from 20 patients performed a year after transplantation revealed no alarming changes [146]. Additionally, a Canadian team examined clinical trials that utilized BM-MSCs. Following a comprehensive examination of 36 studies, they concluded that there was no connection between the application of MSCs and tumorigenic potential, and no significant adverse effects of the treatment were noted [147].

In a recent meta-analysis conducted by Wang and colleagues, a total of 62 randomized clinical trials involving 3,546 people diagnosed with different diseases and treated with intravenous or local implantation, as opposed to receiving no treatment or a placebo, were included. The combined study showed that the administration of MSCs was strongly linked to temporary fever, adverse events at the administration site, constipation, fatigue, and sleeplessness. In conclusion, when compared to alternative placebo modalities, MSC administration has proved safe across a variety of demographics [148].

Clinical studies on MSC treatment have investigated early benefits and hazards, but long-term consequences remain unknown. It is unclear whether therapy advantages are temporary, if additional doses are required, or if the illness worsens. Therefore, it will be necessary to conduct more extensive research and observations on the safety of MSC treatments.

## 7. Conclusions

The emerging research on the role of immune responses in neuroinflammatory diseases such as MS highlights a complex interplay between various T cell subsets, including Th1, Th17, Th2, and Treg cells, as well as the involvement of B cells and CD8^+^ T cells. The promising results from preclinical studies underscore the potential of MSCs in modulating these immune responses and providing neuroprotection. MSCs, with their ability to interact with various immune cells and release anti-inflammatory cytokines, offer a novel therapeutic approach for MS. Their capacity to shift the balance from proinflammatory Th1 and Th17 responses toward more anti-inflammatory Th2 and Treg responses is particularly noteworthy. Clinical trials have shown that MSC therapy is generally safe and holds promise with respect to the improvement of patient outcomes, although further research is needed to optimize treatment protocols and address challenges such as dose determination and potential transplant rejection. The ongoing exploration of MSCs as a treatment modality for MS presents a significant opportunity to develop more effective therapies for this debilitating condition. Future research should focus on refining MSC characterization, understanding their interactions with the immune system, and establishing robust clinical guidelines to fully harness their therapeutic potential.

## Figures and Tables

**Figure 1 cells-13-01556-f001:**
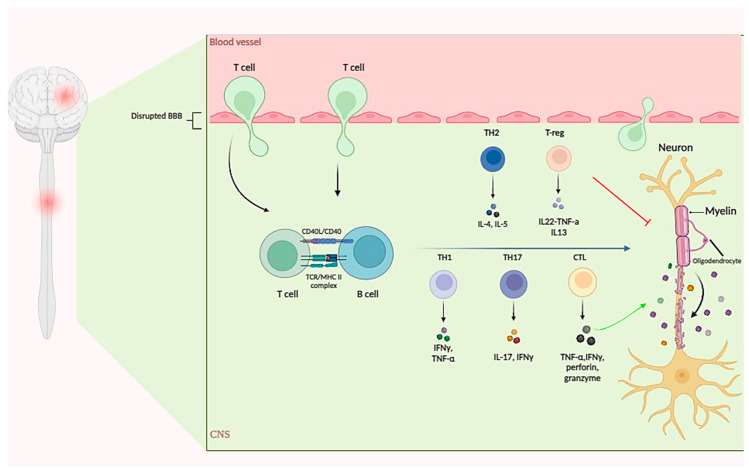
Pathophysiology of multiple sclerosis (MS). MS is an autoimmune neuroinflammatory disease marked by the breakdown of immunological self-tolerance and activation of both innate and adaptive immune systems. Key players include myelin-specific T cells, particularly Th1 and Th17 cells, which drive CNS inflammation and demyelination. Th1 cells secrete proinflammatory cytokines like IFN-γ and TNF-α. Th17 cells produce IL-17 and other cytokines, contributing to blood–brain barrier disruption and CNS lesions. Regulatory T cells (Treg) and Th2 cells counteract inflammation, promoting symptom improvement. Th1 and Th17 cells play complementary roles in MS pathogenesis, while Th2 cells, induced by IL-4, produce anti-inflammatory cytokines (IL-4, IL-10), reducing inflammation and improving symptoms. CD8^+^ T cells (CTLs) predominate in CNS lesions, secreting proinflammatory cytokines (TNFα, IFN-γ) and exhibiting cytotoxicity via perforin and granzyme. They can also produce IL-10, suggesting a regulatory role. B cells contribute to MS pathogenesis through antigen presentation. BBB: blood–brain barrier; TH: T helper cell; TCR: T cell receptor; MHC: major histocompatibility complex; CTL: cytotoxic T lymphocyte; IL: interleukin; TNF: tumor necrosis factor; IFN: interferon.

**Figure 2 cells-13-01556-f002:**
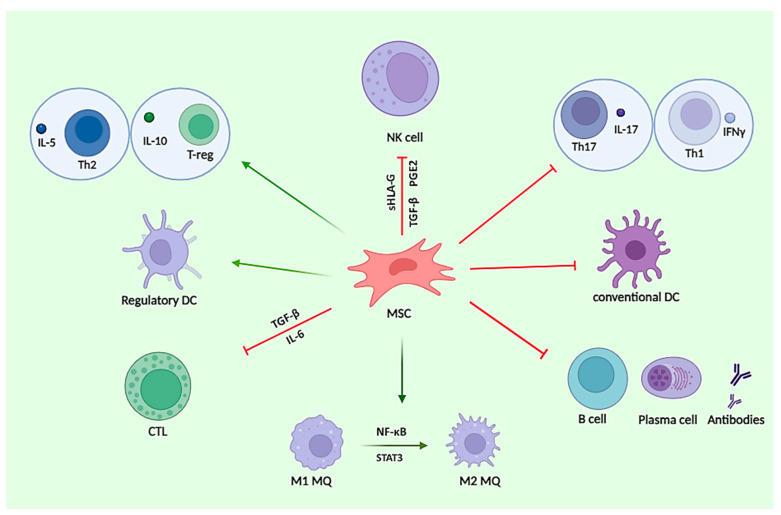
The interplay between MSCs and immune cells. MSCs modulate immune responses by interacting with T, B, NK cells, and DCs. They promote anti-inflammatory phenotypes (Th2, Treg) and inhibit proinflammatory phenotypes (Th1, Th17). MSCs suppress cytotoxic T cell (CTL) proliferation and function, inhibiting B cell differentiation and antibody production. They also modulate dendritic cell (DC) maturation and function, polarize macrophages from the M1 to the M2 phenotype, and suppress NK cell activity through cytokines. NK: natural killer cells; IL: interleukin; IFN: interferon; MQ: macrophage; CTL: cytotoxic T lymphocyte; T-reg: regulatory T cell; Th: T helper cell; TGF: transforming growth factor; PGE: prostaglandin E; NF-κB: nuclear factor kappa-light-chain-enhancer of activated B cells; STAT3: signal transducer and activator of transcription 3; HLA: human leukocyte antigen.

**Figure 3 cells-13-01556-f003:**
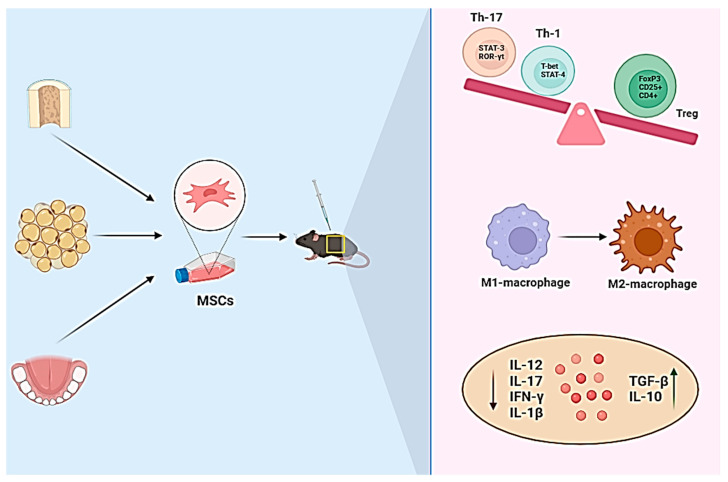
The therapeutic effects of mesenchymal stem cells (MSCs) in the experimental autoimmune encephalomyelitis (EAE) model. The immune modulatory signs that have been observed in EAE animal models that received MSCs derived from various tissues, including adipose, bone marrow, and dental, include an increase in CD4^+^ CD25^+^ FoxP3^+^Treg differentiation from naïve T cell against a decrease in Th1 and Th17 induction. This has led to a rise in the production of anti-inflammatory cytokines like TGF-β and IL-10 and a reduction in the production of proinflammatory cytokines like IL-17, IL-1β, and IFN-γ. Furthermore, MSCs can reduce TNF-α production (not shown in the image) by causing M1 macrophages to polarize into M2 macrophages, alleviating EAE.

**Table 1 cells-13-01556-t001:** Summary of clinical trial information from clinicaltrials.gov regarding the treatment of MS patients through MSC transplantation.

NCT	MSCs	Phase	Enrollment	Summary
NCT02239393	Autologous	2	31	Efficacy is assessed by counting the number of GELs on MRI scans.
NCT04823000	Autologous	1&2	24	Evaluates the safety and tolerability of MSCs repeated treatment.
NCT01854957	Autologous	1&2	20	Efficacy and safety are assessed by counting the number of GELs on MRI scans.
NCT04749667	Autologous	1&2	18	Efficacy is assessed by measuring neurophysiological parameters.
NCT01228266	Autologous	2	9	Efficacy is assessed by clinical variables, MRI, OCT, immunological analysis, and quality of life scales.
NCT03778333	Autologous	1	7	Evaluates changes in EDSS.
NCT00813969	Autologous	1	24	Evaluates safety and tolerability by measuring the number of Gd-enhancing brain MRI lesions.
NCT01377870	Autologous	1&2	22	Efficacy is assessed by MRI, quality of life scales, and RAO test.
NCT03799718	Autologous	2	23	Assesses safety and efficacy using the T25FW and 9-HPT tests.
NCT00395200	Autologous-BM	1&2	10	Assessment of visual function following injection.
NCT00781872	Autologous-BM	1&2	24	Evaluates changes in EDSS, the proportion of T-regs and activated cells, and the proliferation ability of lymphocytes.
NCT02403947	Autologous-BM	1&2	1	Evaluates safety by monitoring for adverse effects and efficacy through MRI of GELs.
NCT01895439	Autologous-BM	1&2	13	Evaluates safety by monitoring for adverse effects and efficacy through MRI and ophthalmological tests.
NCT01745783	Autologous-BM	1&2	26	Evaluates safety by monitoring for adverse effects and efficacy through MRI tests.
NCT02495766	Autologous-BM	1&2	8	Evaluates safety and efficacy by measuring the number of Gd-enhancing brain MRI lesions and EDSS changes.
NCT02035514	Autologous-BM	1&2	9	Evaluates safety by monitoring for adverse effects and efficacy through MRI tests.
NCT02326935	Autologous-AD	1	2	Evaluates safety by monitoring for adverse effects and efficacy through MSIS tests.
NCT01730547	Autologous-AD	1&2	2	Evaluates safety by monitoring for adverse effects and efficacy through MRI and clinical tests.
NCT05116540	Autologous-AD	2	24	Evaluates changes in EDSS and quality of life.
NCT01056471	Autologous-AD	1&2	30	Efficacy is assessed by clinical variables, MRI, neurophysiological and immunological analysis, and quality of life scales.
NCT01933802	Autologous-NP	1	20	Evaluates safety and efficiency by experimental tests like MRI.
NCT06360861	Allogenic-UC	1	5	Evaluates changes in EDSS.
NCT03326505	Allogenic-UC	1&2	60	Safety and efficacy assessment pre- and post-treatment.
NCT01364246	Allogenic-UC	1&2	20	Evaluates safety and efficiency by experimental tests like MRI.
NCT05532943	Allogenic-UC	1&2	41	Evaluates changes in EDSS and quality of life.
NCT05003388	Allogenic-UC	1	15	Evaluates safety by monitoring for adverse effects and efficacy through EDSS tests.
NCT02587715	Allogenic-UC	1&2	69	Evaluates efficacy through EDSS and MRI tests.
NCT02418325	Allogenic-UC	1&2	69	Evaluates efficacy through EDSS and MRI tests.
NCT04956744	hESC	1	30	Assesses the safety and tolerability by monitoring for any adverse effects.

BM: bone marrow, AD: adipose, UC: umbilical cord, hESC: human embryonic stem cells, GEL: gadolinium-enhancing lesions, MRI: magnetic resonance imaging, OCT: optical coherence tomography, Gd: gadolinium, RAO: right anterior oblique, T25FW: timed 25-Foot walk, 9-HPT: nine hole peg test, MSIS: multiple sclerosis impact scale, T-reg: regulatory T cell.

## Data Availability

No new data were created or analyzed in this study. Data sharing is not applicable to this article.

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
