# Peer review of "The Role of Mesenchymal Stem Cells in Modulating Adaptive Immune Responses in Multiple Sclerosis"

_cells, 2024, doi:10.3390/cells13181556_

Round 1

Reviewer 1 Report

Comments and Suggestions for Authors

The review is very compelling concerning the role of mesenchymal stem cells (MSC) in modulating adaptive immune responses in multiple sclerosis (MS). Conventional treatments for MS focus on symptom management rather than addressing the disease’s root cause as MSC therapy is able to do. Detailed immune mechanisms of MSC action as well as results of in vivo model research are presented. Finally, a summary of clinical trial information is shown and limitations, challenges and future directions of MSC treatment are discussed.

Text in all of three figures is poorly readable and must be improved.

Author Response

Reviewer's comments

The review is very compelling concerning the role of mesenchymal stem cells (MSC) in modulating adaptive immune responses in multiple sclerosis (MS). Conventional treatments for MS focus on symptom management rather than addressing the disease’s root cause as MSC therapy is able to do. Detailed immune mechanisms of MSC action as well as results of in vivo model research are presented. Finally, a summary of clinical trial information is shown and limitations, challenges and future directions of MSC treatment are discussed.

Text in all of three figures is poorly readable and must be improved.

Response:

We thank the reviewer for the positive evaluation and constructive feedback. We have now updated the Figures as suggested

Reviewer 2 Report

Comments and Suggestions for Authors

This review is extensive and thorough. Others can benefit by having this literature review to use as a

reference for compiling many concepts. The graphics do help in illustrating the main concepts. The section on Challenges and Future Directions is key to helping other to know where a focus is needed to in moving forward with such MSC related therapies. Indeed, there is still more to be investigated.

There are just some thoughts that come to mind that maybe the authors might want to expand on but not necessary for this review.

1. The authors reference (# 77) “CTL effectors cells by the secretion of suppressive factors (77).”

Are there any known factors which are from supernatant of the harvested from MSC cultures which the authors might mention?

2. It is interesting that the drug (K channel blocker 4-aminopyridine (4-AP) is used in MS and

maybe its actions are not really targeted for neurons but on immune cells such as TEM functions which release of the destructive protease granzyme B. Could maybe blocking the K channels on immune cells be a potential reason for the slight therapeutic actions of 4-AP.

Author Response

Reviewer's comments

This review is extensive and thorough. Others can benefit by having this literature review to use as a reference for compiling many concepts. The graphics do help in illustrating the main concepts. The section on Challenges and Future Directions is key to helping other to know where a focus is needed to in moving forward with such MSC related therapies. Indeed, there is still more to be investigated.

There are just some thoughts that come to mind that maybe the authors might want to expand on but not necessary for this review.

Response. 

We thank the reviewer for the comprehensive and thoughtful review. 

Question 1.

  1. The authors reference (# 77) “CTL effectors cells by the secretion of suppressive factors (77).”

Are there any known factors which are from supernatant of the harvested from MSC cultures which the authors might mention?

Answer 1.

Indeed, as highlighted in the manuscript, from lines 263 to 268, as well as 276 to 287, a range of factors released by mesenchymal stem cells (MSCs) plays a pivotal role in immune suppression. Notably, Figure 2 visually underscores these factors, including TGF-β, IL-6, PGE-2, sHLA-G, and Indoleamine 2,3-dioxygenase (IDO) (1-4).

Question 2.

  1. It is interesting that the drug (K channel blocker 4-aminopyridine (4-AP) is used in MS and maybe its actions are not really targeted for neurons but on immune cells such as TEM functions which release of the destructive protease granzyme B. Could maybe blocking the K channels on immune cells be a potential reason for the slight therapeutic actions of 4-AP.

Answer 2. 

While 4-aminopyridine (4-AP) is primarily associated with neuronal function, exploring its effects beyond neurons opens up intriguing possibilities. 

Neuronal Focus vs. Immune Impact:

Traditionally, 4-AP has been used to enhance axonal conduction in demyelinating disorders like multiple sclerosis (MS). Its primary target has indeed been neuronal potassium channels. However, your suggestion that it might impact immune cells—specifically T effector memory (TEM) cells—is fascinating. These immune cells play a critical role in MS pathogenesis.

Potential Mechanisms

TEM Cells and Granzyme B: TEM cells are memory T cells that rapidly respond to antigen re-exposure. They can release granzyme B, a protease involved in cytotoxicity. K+ Channels on Immune Cells: Blocking K+ channels on immune cells could alter their activation and function. If 4-AP affects these channels, it might influence immune responses.

Experimental Evidence:

Studies have explored blocking voltage-gated K+ channels in immune cells. For instance: In experimental autoimmune encephalomyelitis (EAE), an MS model, blocking these channels improved outcomes (5). Inhibiting K+ channels could potentially modulate immune cell behavior (6).

Therapeutic Implications:

The slight therapeutic effects of 4-AP in MS might indeed involve immune modulation. Investigating the precise interactions between 4-AP and immune cells, including granzyme B release, could yield valuable insights. Perhaps K+ channels serve as targets for specific immunomodulation.

  1. Fletcher JM, Lalor SJ, Sweeney CM, Tubridy N, Mills KH. T cells in multiple sclerosis and experimental autoimmune encephalomyelitis. Clin Exp Immunol. 2010;162(1):1-11.
  2. Lee HJ, Jung H, Kim DK. IDO and CD40 May Be Key Molecules for Immunomodulatory Capacity of the Primed Tonsil-Derived Mesenchymal Stem Cells. Int J Mol Sci. 2021;22(11).
  3. Manel N, Unutmaz D, Littman DR. The differentiation of human T(H)-17 cells requires transforming growth factor-beta and induction of the nuclear receptor RORgammat. Nat Immunol. 2008;9(6):641-9.
  4. Sotiropoulou PA, Perez SA, Gritzapis AD, Baxevanis CN, Papamichail M. Interactions between human mesenchymal stem cells and natural killer cells. Stem Cells. 2006;24(1):74-85.
  5. Beeton C, Barbaria J, Giraud P, Devaux J, Benoliel AM, Gola M, et al. Selective blocking of voltage-gated K+ channels improves experimental autoimmune encephalomyelitis and inhibits T cell activation. J Immunol. 2001;166(2):936-44.
  6. Chandy KG, Wulff H, Beeton C, Pennington M, Gutman GA, Cahalan MD. K+ channels as targets for specific immunomodulation. Trends Pharmacol Sci. 2004;25(5):280-9.